# Serum Zinc and Long-Term Prognosis after Acute Traumatic Brain Injury with Intracranial Injury: A Multicenter Prospective Study [note 1]

**DOI:** 10.3390/jcm11216496

**Published:** 2022-11-01

**Authors:** Ki Hong Kim, Young Sun Ro, Hanna Yoon, Stephen Gyung Won Lee, Eujene Jung, Sung Bae Moon, Gwan Jin Park, Sang Do Shin

**Affiliations:** 1Department of Emergency Medicine, Seoul National University Hospital, Seoul 03080, Korea; 2Laboratory of Emergency Medical Services, Seoul National University Hospital Biomedical Research Institute, Seoul 03080, Korea; 3Department of Emergency Medicine, Seoul National University Boramae Medical Center, Seoul 07061, Korea; 4Department of Emergency Medicine, Chonnam National University Hospital, Gwangju 61469, Korea; 5Department of Emergency Medicine, Kyungpook National University School of Medicine and Kyungpook National University Hospital, Daegu 41404, Korea; 6Department of Emergency Medicine, Chungbuk National University Hospital, Cheongju 28644, Korea

**Keywords:** traumatic brain injury, prognosis, zinc

## Abstract

Serum zinc levels in the acute stages after traumatic brain injury (TBI) may be capable of predicting cinical and functional prognoses. This study aimed to evaluate the association between serum zinc levels and long-term survival and neurological outcomes in TBI patients with intracranial injury. This multicenter prospective cohort study enrolled adult TBI patients with intracranial injury who visited emergency departments between December 2018 and June 2020. Serum zinc levels drawn within 24 h after injury were categorized into four groups: low (<80.0 mcg/dL), low–normal (80.0–100.0 mcg/dL), high–normal (100.1–120.0 mcg/dL), and high (>120.0 mcg/dL). The study outcomes were 6-month mortality and disability (Glasgow Outcome Scale, 1–3). A multilevel multivariable logistic regression analysis was conducted to estimate associations between serum zinc and study outcomes. From the eligible TBI patients (N = 487), the median (interquartile range) serum zinc level was 112.0 mcg/dL (95.0–142.0). Six-month mortality and disability were 21.1% (103/487) and 29.6% (144/487), respectively. Compared to the high–normal zinc group, there were significant associations with 6-month mortality and disability observed in the low zinc group (aORs (95% CIs): 1.91 (1.60–2.28) and 1.95 (1.62–2.36) for the low group; 1.14 (0.67–1.94) and 1.15 (0.91–1.46) for the low–normal group; and 0.72 (0.44–1.16) and 0.88 (0.61–1.27) for the high group, respectively). Among the 122 TBI patients with diabetes mellitus, the low zinc group showed a higher incidence of 6-month mortality (aOR (95% CI): 9.13 (4.01–20.81)) compared to the high–normal zinc group. Moreover, the low and low–normal groups had higher odds for 6-month disability (aORs (95% CIs): 6.63 (3.61–12.15) for the low group and 2.37 (1.38–4.07) for the low–normal group). Serum zinc deficiency is associated with a higher incidence of 6-month mortality and disability after injury for TBI patients with intracranial injury.

## 1. Introduction

Traumatic brain injury (TBI) is the leading cause of all injury-related deaths and a major public health concern with high mortality and morbidity worldwide [1]. A large medical and economic burden over a long period of time also has a large social impact [2,3]. Accurate predictive models for the diagnosis and prognosis of intracranial injury have been necessary for an effective therapeutic strategy to improve clinical outcomes and decrease disease burden of TBI [4]. There have been many approaches to finding cellular and biochemical biomarkers predicting the prognosis of TBI [5,6].

Zinc is a trace metal that is essential for normal development of the central nervous system in the early neonatal periods and the maintenance of brain function in adults [7]. It is an important element present in the hippocampus, neo-cortex, amygdala, olfactory bulbs, and hypothalamus [8,9]. Zinc is known to play an important role in neurological recovery through the regulation of oxidative stress that causes secondary injury [10], and changes in brain and neuronal zinc due to trauma, stroke, and seizures have been associated with neuronal damage and death [11]. With induction of progenitor cell proliferation and neurogenesis, zinc is also essential in the modulation of hippocampal neurogenesis after TBI [12]. Chelation of free zinc to the brain in animal TBI models showed evidence of a reduced injured neurons and a protective effect [13,14]. It has been suggested that zinc deficiency should be prevented to optimize the neurological recovery potential in acute care after TBI [15].

Zinc may be considered as a surrogate marker as serum zinc levels decrease and urine zinc excretion increases in proportion to the severity of TBI [16]. Moreover, it has been suggested that zinc supplementation given as a treatment after TBI can effectively reduce cognitive impairment and depression associated with TBI [17,18,19]. Serum zinc levels in the early stages of severe TBI may be important in predicting neurological outcomes and determining zinc supplementation, but this has not been well studied. 

This prospective study aimed to evaluate the association between serum zinc level after TBI and long- and short-term neurological outcomes of TBI patients with intracranial hemorrhage or diffuse axonal injury who are visiting the emergency department (ED), and to evaluate whether this association is dependent on diabetes mellitus, which is known to delay zinc metabolism [20]. We hypothesized that serum zinc deficiency at the acute stage after TBI would be associated with long- and short-term poor neurological outcomes, and that TBI patients diagnosed with diabetes mellitus would have a stronger such association. 

## 2. Materials and Methods

### 2.1. Study Design, Setting, and Data Sources 

This was a multicenter prospective cohort study conducted in five participating academic hospitals in Korea based on the Pan-Asian Trauma Outcome Study for Traumatic Brain Injury (PATOS-TBI) registry (ClinicalTrials.gov, ID: NCT04718935). 

The purpose of the PATOS-TBI study is to clarify nutritional and metabolic biomarkers related to the prognoses of TBI. This study prospectively enrolled adult TBI patients with intracranial injury, including diffuse axial injury and intracranial hemorrhage, transported by emergency medical services (EMS) to participating EDs within 72 h after injury. Patients diagnosed with intracranial injury based on radiological tests in the ED through computer tomography (CT) or magnetic resonance imaging (MRI) were included. Informed consent was required and obtained from patients, or caregivers in cases of unconsciousness [21].

The PATOS-TBI study registry includes demographics; injury characteristics; and clinical findings, including vital signs upon ED arrival, neurological examinations, findings of laboratory tests and radiological tests, diagnoses and in-hospital treatment, and survival and functional outcomes at time of hospital discharge and follow up. Upon confirmation of intracranial injury and consent to enrolling in the study, blood samples for biomarkers were obtained in the ED. Long-term functional outcomes were investigated and collected by telephone interviews at 1 and 6 months after injury. For quality assurance of the PATOS-TBI registry, research coordinators were required to receive education and training prior to participation in the study and periodically during the study period. Quality control of entered data was performed by the PATOS-TBI data quality control team on a monthly basis. 

### 2.2. Study Population

Adult TBI patients over 18 years of age who visited participating EDs between December 2018 and June 2020 were enrolled. Patients with unknown information on long-term functional outcomes or serum zinc levels and patients whose blood samples were drawn 24 h after injury were excluded. 

### 2.3. Outcome Measures 

The primary outcomes were mortality and disability at 6 months after injury. Secondary outcomes were mortality and disability at 1 month after injury. The Glasgow Outcome Scale (GOS) was scored from 1 to 5 as follows: 1 (death), 2 (vegetative state), 3 (severe disability), 4 (moderate disability), and 5 (good recovery). Disability after injury was defined as a GOS score of 1, 2, and 3. Research coordinators at each of the five participating hospitals conducted interviews with structured questions to assess the functional outcomes (GOS, EQ-5D, and reason of mortality) of patients at 1 and 6 months after injury.

### 2.4. Analysis of Serum Zinc 

All patients diagnosed with TBI with intracranial injury immediately had their blood samples (24 mL) drawn from the peripheral veins. The blood samples were centrifuged (10 min at 3000 rpm, room temperature) and refrigerated at −20°C within 60 min after being drawn. After periods of patient enrollment, zinc assays were conducted for all eligible patients. Inductively coupled plasma mass spectrometry (ICP-MS) was used to measure zinc, including free zinc and the protein-binding form, in metal-free serum. ICP-MS Autosampler (ASX-520), a recirculating chiller (G3292A), and a mass spectrometer (7900 ICP-MS) were used (Agilent Technologies, Palo Alto, CA, USA).

### 2.5. Measurements and Variables 

The main exposure variable was serum zinc levels, categorized into four groups considering the normal range for serum zinc level (80.0–120.0 mcg/dL): low (0.0~79.9 mcg/dL), low–normal (80.0~100.0 mcg/dL), high–normal (100.1~120.0 mcg/dL), and high (120.1~320.0 mcg/dL) group [22,23].

Demographics and clinical findings at the ED were collected, including age, sex, education, pre-injury disability (GOS 1–4), comorbidities (hypertension, diabetes mellitus, chronic liver disease, hemodialysis, coagulopathy, and antiplatelet medication or anticoagulation), mechanism of injury (road traffic injury, fall, and others), prehospital alertness, vital signs and mentality at the ED triage, initial laboratory findings (hemoglobin, platelet, and prothrombin time international ratio), final diagnosis, injury severity score, in-hospital management (transfusion and operation), and hospital outcomes. 

### 2.6. Statistical Analysis

Categorical variables were set as counts and percentages using Chi-squared tests. Continuous variables were set as the median and interquartile range (IQR). A *p*-value under 0.05 was considered as the measure of statistical significance with Bonferroni adjustment for multiple comparisons. To determine associations between the study group and the study outcomes, adjusted odds ratios (aORs) with 95% confidence intervals (95% CIs) were calculated, using the generalized linear mixed models for multilevel logistic regression analyses, since the hospital data were nested. The potential confounders based on previous studies were adjusted, including age groups, sex, education (high school graduate), pre-injury disability, and comorbidities (hypertension, diabetes mellitus, and chronic liver disease). 

Sensitivity analyses were conducted according to time from injury to blood sample collection (18, 12, and 6 h), since serum zinc levels could have been altered during that time period, from urinary secretion or release from liver after injury. Moreover, subgroup analyses for TBI patients previously diagnosed with diabetes mellitus before injury were conducted to evaluate the magnitude of association between serum zinc level and study outcomes in high-risk populations. 

All statistical analyses were conducted with SAS software, version 9.4 (SAS Institute Inc., Cary, NC, USA). 

## 3. Results

Of the 606 TBI patients with intracranial hemorrhage or diffuse axonal injury who visited EDs, 487 eligible patients were enrolled after excluding patients with unknown information on 6-month outcomes (*n* = 62) and serum zinc levels (*n* = 2), and whose time from injury to blood sample collection exceeded 24 h (*n* = 55). 

Table 1 presents the characteristics of the study participants according to serum zinc levels. The median (interquartile range) serum zinc level was 112.0 mcg/dL (95.0–142.0); 43 patients (8.8%) were included in the low group, 114 patients (23.4%) were included in the low–normal group, 136 patients (27.9%) were included in the high–normal group, and 194 patients (39.8%) were included in the high group.

The proportion of 6-month mortality was 21.1% (103/487) in the study population: 34.9% (15/43) in the low group, 25.4% (38/114) in the low–normal group, 21.3% (29/136) in the high–normal group, and 15.5% (30/194) in the high group (*p*-value = 0.02). Proportion of disability at 6 months after injury was 44.2% in the low group, 32.5% in the low–normal group, 28.7% in the high–normal group, and 25.3% in the high group (*p*-value = 0.08). 

In the multilevel multivariable logistic regression analysis, there was a significant association between the serum zinc levels and survival and functional outcomes after TBI with intracranial injury. Compared to the high–normal zinc group, the low group had higher odds of 6-month mortality (aOR (95% CI): 1.91 (1.60–2.28)) and 6-month disability (aOR (95% CI): 1.95 (1.62–2.36)). In terms of 1-month mortality and disability, the low group had higher odds (aORs (95% CIs): 2.52 (1.92–3.30) for 1-month mortality and 1.90 (1.45–2.49) for 1-month disability) than the high–normal group. The low–normal and high zinc group had no significant differences for study outcomes compared to the high–normal group (Table 2). Independent predictive performance of serum zinc levels on long- and short-term neurological prognoses was evaluated, and sensitivity for 6-month mortality for low serum zinc level (~79.9 mcg/dL) was 0.9271 (Appendix A).

The sensitivity analysis—intended to assess whether associations between serum zinc levels and study outcomes were maintained in patients for whom blood samples were drawn within 18 (N = 475), 12 (N = 456), and 6 (N = 414) hours after injury—shows similar trends to the aforementioned associations (Table 3). Compared to the high–normal zinc group, the low group had significantly higher odds for study outcomes, with aORs (95% CIs) for 6-month mortality and disability of 1.76 (1.47–2.10) and 1.77 (1.48–2.12) for within 18 h, 1.80 (1.49–2.17) and 1.80 (1.47–2.21) for within 12 h, and 1.90 (1.62–2.24) and 1.83 (1.43–2.35) for within 6 h, respectively.

Among the 122 TBI patients previously diagnosed with diabetes mellitus, the proportions of 6-month mortality and disability were 19.7% (24/144) and 27.9% (34/122), respectively. The low zinc group showed higher odds for 6-month and 1-month mortality compared to the high–normal group (aORs (95% CIs): 9.13 (4.01–20.81) for 6-month and 10.71 (3.67–31.28) for 1-month mortality). In terms of disability, compared with the high–normal zinc group, the low and the low–normal groups had higher odds for 6-month disability (aORs (95% CIs): 6.63 (3.61–12.15) for the low group and 2.37 (1.38–4.07) for the low–normal group) as well as 1-month disability (aORs (95% CIs): 3.76 (1.67–8.50) for the low group and 1.87 (1.11–3.13) for the low–normal group) (Table 4).

## 4. Discussion

Using the multicenter prospective PATOS-TBI registry, this study found that serum zinc levels in the acute stage after injury has associations with long-term survival and neurological outcomes of TBI patients with intracranial hemorrhage or diffuse axonal injury (aORs (95% CIs): 1.91 (1.60–2.28) for 6-month mortality and 1.95 (1.62–2.36) for 6-month disability). In the sensitivity analysis regarding time from injury to blood sample collection, such an association was maintained for TBI patient groups with blood samples drawn within 18 h, 12 h, and 6 h. The association between serum zinc level and long-term survival and neurological outcomes was shown to be stronger, especially for TBI patients previously diagnosed with diabetes mellitus. 

Zinc is known as an essential component for neurogenesis from early neonatal periods to adulthood [7]. Regarding TBI, zinc is also essential in the modulation of neurogenesis after injury [12]; in the regulation of oxidative stress that causes secondary injury [10]; and in improving the balance of proteins, such as neurotrophins, which modulate neurogenesis [17,24,25,26,27]. In recent animal studies, zinc supplementation showed improvements in spatial learning and memory and hippocampal neuronal precursor cell regenesis [19,28]. Neuronal precursor cell proliferation and neurogenesis were found in the hippocampus after TBI [28], while zinc administration shortly after TBI demonstrated enhanced protein synthesis and improved consciousness [27]. The neuroprotection during the acute phase after TBI and neuro-regeneration in the recovery phase are crucial for patients’ neurological and functional outcomes. This may provide a theoretical basis for supplementation before and after injury in patients with serum zinc deficiency.

Alterations in free zinc ion homeostasis, such as accumulation in neurons, are known to impair advanced brain functions, including memory and cognition [29]. S100B, a zinc-binding protein, affects local zinc concentrations by scavenging free zinc ions from extracellular release and regulates hyperexcitation of glutamatergic neurons in the central nervous system [30]. It has been demonstrated that zinc-bound S100B has the potential to modulate excitotoxicity by altering calcium signaling, which could be reflected in serum zinc levels [31]. Moreover, zinc plays a role in regulating tau phosphorylation, which may contribute to pathological structural changes in the TBI, and zinc supplementation is suggested to decrease tau phosphorylation by negatively regulating the amount of GSK-3-beta [32,33]. Considering that serum zinc level is physiologically decreased during the acute phase of TBI, the failure of the tau phosphorylation role in response to dynamic changes in zinc amount may be associated with poor prognosis.

Diabetes mellitus has been known to have an association with zinc dysregulation [34]. In this study, patients previously diagnosed with diabetes mellitus showed a higher probability of grave neurological outcomes, even within the low–normal range compared to high–normal (aORs (95% CIs) for 6-month disability: 6.63 (3.61–12.15) for the low group and 2.37 (1.38–4.07) for the low–normal group). Serum zinc levels in patients with type 1 and type 2 diabetes are known to be lower than in non-diabetic patients [35,36]. Zinc supplements are recommended for diabetic patients due to their antioxidant effects on multiple organs including prevention of coronary heart disease [37]. Since zinc would be released from the liver and be excreted through the kidneys after TBI [16], it can be assumed that zinc levels in the ED indicated not only general serum zinc level, but total storage of zinc. Considering this association, serum zinc level of patients with diabetes mellitus should be managed to improve neurological outcomes after TBI with intracranial injury. 

In this study, zinc deficiency was associated with long-term mortality and poor functional recovery in TBI patients with intracranial injury. Especially in patients with diabetes mellitus, the low–normal zinc level was associated with long- and short-term disability. Nutritional health, including serum zinc, is associated with clinical recovery after TBI. Based on the results, it is reasonable to consider initial serum zinc level as a biomarker to rapidly determine long-term mortality and neurological outcomes in TBI with intracranial injury. Further clinical studies on the effects of zinc supplementation after TBI and appropriate doses of zinc administration for TBI treatment should be conducted to develop strategies to improve functional and cognitive recovery after injury, especially for high-risk populations such as patients with diabetes mellitus or chronic liver disease.

### Limitations 

This study has several limitations. First, the kinetics of serum zinc levels in relation to elapsed time from TBI is not yet understood, and could have affected the results. Blood samples in this study were collected only at the time of ED arrival, and even though sensitivity analysis was conducted to ensure homogeneity of time from injury to blood sample, caution should be exercised when interpreting the study results. Second, there were no available data on the history of zinc deficiency and nutritional status of eligible patients before TBI, both of which would be related to serum zinc levels after injury and neurological outcomes after TBI. In addition, there was no information about whether zinc was supplemented after TBI. Future studies are needed to determine whether associations between serum zinc levels and severe TBI outcomes vary according to these differences. Third, patients with unknown GOS scores at 6 months were excluded, and thus selection bias may arise. Fourth, since the study population was not limited to isolated TBI patients, there is a possibility that critical injuries in anatomical regions other than the brain may have affected the prognosis of the patients. Although head AIS was adjusted for in the final statistical model, uncontrolled bias may occur. Finally, the study design was a multicenter prospective cohort study, not a randomized controlled trial. There may have been significant potential unmeasured biases that were not controlled. 

## 5. Conclusions

Serum zinc deficiency is associated with long- and short-term mortality and poor functional recovery for TBI patients with intracranial hemorrhage and diffuse axonal injury. Furthermore, this association is stronger in TBI patients previously diagnosed with diabetes mellitus.

## Figures and Tables

**Table 1 jcm-11-06496-t001:** Demographics of study population according to serum zinc level.

	Total	Serum Zinc Group				
		Low, 0.0~79.9	Low–Normal, 80.0~100.0	High–Normal, 100.1~120.0	High 120.1~320.0	*p*-Value
	N (%)	N (%)	N (%)	N (%)	N (%)	
Total	487	43	114	136	194	
Serum zinc levels [mcg/dl], median (IQR)	112.0 (95.0–142.0)	69.4 (65.2–74.6)	91.2 (86.3–96.1)	109.2 (105.9–114.4)	147.7 (134.6–175.2)	
Age, year, median (IQR)	67 (55–77)	64 (51–77)	71 (57–77)	68 (58–78.5)	64 (53–74)	0.09
Male sex	334 (68.6)	32 (74.4)	79 (69.3)	88 (64.7)	135 (69.6)	0.63
Education, >12 years	218 (44.8)	19 (44.2)	53 (46.5)	53 (39.0)	93 (47.9)	0.43
Pre-injury disability, GOS 1–4	9 (1.8)	–	2 (1.8)	4 (2.9)	3 (1.5)	0.75
Comorbidities						
Hypertension	182 (37.4)	14 (32.6)	39 (34.2)	54 (39.7)	75 (38.7)	0.71
Diabetes mellitus	122 (25.1)	14 (32.6)	28 (24.6)	43 (31.6)	37 (19.1)	0.04
Chronic liver disease	22 (4.5)	3 (7.0)	4 (3.5)	6 (4.4)	9 (4.6)	0.83
Hemodialysis	18 (3.7)	–	7 (6.1)	8 (5.9)	3 (1.5)	0.04
Coagulopathy	11 (2.3)	–	7 (6.1)	1 (0.7)	3 (1.5)	0.01
Antiplatelet or anticoagulation	61 (12.5)	3 (7.0)	17 (14.9)	18 (13.2)	23 (11.9)	0.58
Mechanism of injury						0.02
Road traffic injury	210 (43.1)	18 (41.9)	58 (50.9)	49 (36.0)	85 (43.8)	
Fall	204 (41.9)	12 (27.9)	44 (38.6)	66 (48.5)	82 (42.3)	
Others	73 (15.0)	13 (30.2)	12 (10.5)	21 (15.4)	27 (13.9)	
Prehospital alertness	133 (27.3)	6 (14.0)	27 (23.7)	34 (25.0)	66 (34.0)	0.02
GCS in the ED, median (IQR)	15 (9–15)	12 (4–15) ^b,c^	15 (7–15)	15 (11–15)	15 (10–15)	0.02
Laboratory findings, median (IQR)						
Hemoglobin [g/dL]	12 (10–13)	10 (8.8–11) ^a,b,c^	11 (10–13) ^b,c^	12 (11–13)	13 (11–14)	<0.01
Platelet [10^3^/μL]	194 (159–244)	168 (144–228)	200 (165–242)	189 (152–245)	198 (167–244)	0.07
PT INR	1.0 (0.9–1.1)	1.1 (1–1.2) ^a,b,c^	1.0 (0.9–1.1)	1.0 (0.9–1.1)	1.0 (0.9–1.0)	<0.01
Types of intracranial injury						
Diffuse axonal injury	34 (7.0)	4 (9.3)	6 (5.3)	7 (5.1)	17 (8.8)	0.47
Subdural hemorrhage	359 (73.7)	30 (69.8)	84 (73.7)	99 (72.8)	146 (75.3)	0.89
Epidural hemorrhage	74 (15.2)	8 (18.6)	13 (11.4)	28 (20.6)	25 (12.9)	0.14
Subarachnoid hemorrhage	200 (41.1)	14 (32.6)	51 (44.7)	52 (38.2)	83 (42.8)	0.46
Intracerebral hemorrhage	115 (23.6)	13 (30.2)	33 (28.9)	30 (22.1)	39 (20.1)	0.23
Intraventricular hemorrhage	41 (8.4)	7 (16.3)	7 (6.1)	7 (5.1)	20 (10.3)	0.07
AIS of head injury, 3–6	407 (83.6)	38 (88.4)	100 (87.7)	110 (80.9)	159 (82.0)	0.36
Injury severity score	17 (10–25)	22 (16–27) ^c^	19.5 (13–25) ^c^	16 (10–25)	16 (9–22)	<0.01
Transfusion in the ED	127 (26.1)	21 (48.8) ^b,c^	41 (36.0) ^b,c^	27 (19.9)	38 (19.6)	<0.01
Any operation	138 (28.3)	14 (32.6)	30 (26.3)	42 (30.9)	52 (26.8)	0.74
Outcomes						
ED mortality	15 (3.1)	1 (2.3)	5 (4.4)	3 (2.2)	6 (3.1)	
1-month GOS						0.19
Death	95 (19.5)	15 (34.9)	28 (24.6)	24 (17.6)	28 (14.4)	
Vegetative state	6 (1.2)	–	3 (2.6)	1 (0.7)	2 (1.0)	
Severe disability	45 (9.2)	3 (7.0)	8 (7.0)	14 (10.3)	20 (10.3)	
Moderate disability	38 (7.8)	4 (9.3)	8 (7.0)	9 (6.6)	17 (8.8)	
Good recovery	303 (62.2)	21 (48.8)	67 (58.8)	88 (64.7)	127 (65.5)	
6-month GOS						0.35
Death	103 (21.1)	15 (34.9)	29 (25.4)	29 (21.3)	30 (15.5)	
Vegetative state	8 (1.6)	–	2 (1.8)	2 (1.5)	4 (2.1)	
Severe disability	33 (6.8)	4 (9.3)	6 (5.3)	8 (5.9)	15 (7.7)	
Moderate disability	35 (7.2)	4 (9.3)	8 (7.0)	8 (5.9)	15 (7.7)	
Good recovery	308 (63.2)	20 (46.5)	69 (60.5)	89 (65.4)	130 (67.0)	

Abbreviations: IQR—interquartile range; GOS—Glasgow outcome scale; GCS—Glasgow coma scale; ED—emergency department; PT INR—prothrombin time international normalized ratio; AIS—abbreviated injury scale; ^a^ Bonferroni-adjusted *p*-value < 0.05 compared with the low–normal group; ^b^ Bonferroni-adjusted *p*-value < 0.05 compared with the high–normal group; ^c^ Bonferroni-adjusted *p*-value < 0.05 compared with the high group.

**Table 2 jcm-11-06496-t002:** Multilevel logistic regression analysis for study outcomes according to serum zinc levels.

	Outcome	Unadjusted Model	Model 1	Model 2
	*n*/N (%)	OR (95% CIs)	Adjusted OR (95% CIs)	Adjusted OR (95% CIs)
6-month mortality				
Low, 0.0~79.9 mcg/dL	15/43 (34.9)	1.98 (0.93–4.18)	1.92 (1.63–2.26)	1.91 (1.60–2.28)
Low–normal, 80.0~100.0 mcg/dL	29/114 (25.4)	1.26 (0.70–2.27)	1.27 (0.71–2.25)	1.14 (0.67–1.94)
High–normal, 100.1~120.0 mcg/dL	29/136 (21.3)	1.00	1.00	1.00
High, 120.1~ mcg/dL	30/194 (15.5)	0.68 (0.38–1.19)	0.78 (0.48–1.25)	0.72 (0.44–1.16)
6-month disability, GOS 1–3				
Low, 0.0~79.9 mcg/dL	19/43 (44.2)	1.97 (0.97–4.00)	1.98 (1.66–2.37)	1.95 (1.62–2.36)
Low–normal, 80.0~100.0 mcg/dL	37/114 (32.5)	1.20 (0.70–2.05)	1.23 (0.92–1.63)	1.15 (0.91–1.46)
High–normal, 100.1~120.0 mcg/dL	39/136 (28.7)	1.00	1.00	1.00
High, 120.1~ mcg/dL	49/194 (25.3)	0.84 (0.51–1.38)	0.93 (0.66–1.31)	0.88 (0.61–1.27)
1-month mortality				
Low, 0.0~79.9 mcg/dL	15/43 (34.9)	2.50 (1.16–5.38)	2.56 (2.00–3.29)	2.52 (1.92–3.30)
Low–normal, 80.0~100.0 mcg/dL	28/114 (24.6)	1.52 (0.82–2.81)	1.57 (0.76–3.25)	1.41 (0.74–2.68)
High–normal, 100.1~120.0 mcg/dL	24/136 (17.6)	1.00	1.00	1.00
High, 120.1~ mcg/dL	28/194 (14.4)	0.79 (0.43–1.43)	0.91 (0.50–1.63)	0.84 (0.48–1.48)
1-month disability, GOS 1–3				
Low, 0.0~79.9 mcg/dL	18/43 (41.9)	1.79 (0.88–3.65)	1.94 (1.48–2.54)	1.90 (1.45–2.49)
Low–normal, 80.0~100.0 mcg/dL	39/114 (34.2)	1.29 (0.76–2.21)	1.39 (0.97–1.98)	1.32 (0.93–1.86)
High–normal, 100.1~120.0 mcg/dL	39/136 (28.7)	1.00	1.00	1.00
High, 120.1~ mcg/dL	50/194 (25.8)	0.86 (0.53–1.41)	0.93 (0.56–1.54)	0.92 (0.56–1.51)

Model 1: adjusted for age, sex, education, and pre-injury disability; Model 2: adjusted for age, sex, education, pre-injury disability, injury severity (head AIS), and comorbidities (hypertension, diabetes mellitus, and chronic liver disease). Abbreviations: OR—odds ratio; CI—confidence interval; GOS—Glasgow outcome scale.

**Table 3 jcm-11-06496-t003:** Sensitivity analysis according to time from injury to blood sample collection.

	Time from Injury to Blood Sample Collection		
	0–18 h (N= 475)	0–12 h (N = 456)	0–6 h (N = 414)
	Adjusted OR (95% CI)	Adjusted OR (95% CI)	Adjusted OR (95% CI)
6-month mortality			
Low, 0.0~79.9 mcg/dL	1.76 (1.47–2.10)	1.80 (1.49–2.17)	1.90 (1.62–2.24)
Low–normal, 80.0~100.0 mcg/dL	1.04 (0.62–1.76)	1.01 (0.56–1.82)	1.03 (0.59–1.79)
High–normal, 100.1~120.0 mcg/dL	1.00	1.00	1.00
High, 120.1~ mcg/dL	0.69 (0.42–1.15)	0.74 (0.47–1.16)	0.83 (0.55–1.25)
6-month disability, GOS 1–3			
Low, 0.0~79.9 mcg/dL	1.77 (1.48–2.12)	1.80 (1.47–2.21)	1.83 (1.43–2.35)
Low–normal, 80.0~100.0 mcg/dL	1.05 (0.83–1.33)	1.01 (0.77–1.31)	1.13 (0.88–1.44)
High–normal, 100.1~120.0 mcg/dL	1.00	1.00	1.00
High, 120.1~ mcg/dL	0.84 (0.57–1.25)	0.84 (0.59–1.21)	0.98 (0.70–1.36)
1-month mortality			
Low, 0.0~79.9 mcg/dL	2.32 (1.78–3.03)	2.43 (1.80–3.28)	2.39 (1.85–3.09)
Low–normal, 80.0~100.0 mcg/dL	1.29 (0.68–2.44)	1.28 (0.63–2.60)	1.19 (0.63–2.26)
High–normal, 100.1~120.0 mcg/dL	1.00	1.00	1.00
High, 120.1~ mcg/dL	0.81 (0.44–1.48)	0.88 (0.50–1.53)	0.97 (0.60–1.57)
1-month disability, GOS 1–3			
Low, 0.0~79.9 mcg/dL	1.73 (1.30–2.30)	1.90 (1.56–2.31)	1.81 (1.53–2.13)
Low–normal, 80.0~100.0 mcg/dL	1.20 (0.82–1.76)	1.20 (0.94–1.54)	1.30 (0.98–1.72)
High–normal, 100.1~120.0 mcg/dL	1.00	1.00	1.00
High, 120.1~ mcg/dL	0.88 (0.52–1.49)	0.94 (0.60–1.48)	1.06 (0.67–1.68)

Adjusted for age, sex, education, pre-injury disability, injury severity (head AIS), and comorbidities (hypertension, diabetes mellitus, and chronic liver disease). Abbreviations: OR—odds ratio; CI—confidence interval; GOS—Glasgow outcome scale.

**Table 4 jcm-11-06496-t004:** Subgroup analysis for TBI patients diagnosed with diabetes mellitus (*n* = 122).

	Outcome	Unadjusted Model	Model 1	Model 2
	*n*/N (%)	OR (95% CI)	Adjusted OR (95% CI)	Adjusted OR (95% CI)
6-month mortality				
Low, 0.0~79.9 mcg/dL	6/14 (42.9)	4.63 (1.18–18.12)	8.63 (4.03–18.49)	9.13 (4.01–20.81)
Low–normal, 80.0~100.0 mcg/dL	6/28 (21.4)	1.68 (0.48–5.86)	1.99 (0.94–4.22)	1.98 (0.86–4.53)
High–normal, 100.1~120.0 mcg/dL	6/43 (14.0)	1.00	1.00	1.00
High, 120.1~ mcg/dL	6/37 (16.2)	1.19 (0.35–4.08)	1.51 (0.45–5.14)	1.34 (0.41–4.37)
6-month disability, GOS 1–3				
Low, 0.0~79.9 mcg/dL	6/14 (42.9)	2.48 (0.69–8.84)	5.16 (2.88–9.25)	6.63 (3.61–12.15)
Low–normal, 80.0~100.0 mcg/dL	8/28 (28.6)	1.32 (0.45–3.90)	2.05 (1.13–3.72)	2.37 (1.38–4.07)
High–normal, 100.1~120.0 mcg/dL	10/43 (23.3)	1.00	1.00	1.00
High, 120.1~ mcg/dL	10/37 (27.0)	1.22 (0.44–3.37)	1.85 (0.51–6.75)	1.75 (0.53–5.71)
1-month mortality				
Low, 0.0~79.9 mcg/dL	6/14 (42.9)	5.70 (1.39–23.36)	11.63 (3.89–34.77)	10.71 (3.67–31.28)
Low–normal, 80.0~100.0 mcg/dL	6/28 (21.4)	2.07 (0.57–7.59)	2.60 (0.91–7.46)	2.40 (0.77–7.53)
High–normal, 100.1~120.0 mcg/dL	5/43 (11.6)	1.00	1.00	1.00
High, 120.1~ mcg/dL	6/37 (16.2)	1.47 (0.41–5.28)	1.99 (0.55–7.16)	1.71 (0.47–6.19)
1-month disability, GOS 1–3				
Low, 0.0~79.9 mcg/dL	6/14 (42.9)	1.94 (0.56–6.77)	3.22 (1.33–7.78)	3.76 (1.67–8.50)
Low–normal, 80.0~100.0 mcg/dL	9/28 (32.1)	1.22 (0.43–3.45)	1.75 (0.99–3.09)	1.87 (1.11–3.13)
High–normal, 100.1~120.0 mcg/dL	12/43 (27.9)	1.00	1.00	1.00
High, 120.1~ mcg/dL	10/37 (27.0)	0.96 (0.36–2.56)	1.16 (0.57–2.38)	1.02 (0.49–2.11)

Adjusted for age, sex, education, pre-injury disability, injury severity (head AIS), and comorbidities (hypertension, diabetes mellitus, and chronic liver disease). Abbreviations: OR—odds ratio; CI—confidence interval; GOS—Glasgow outcome scale.

## Data Availability

The data for this study were obtained from the National Research Foundation of Korea (NRF). Restrictions apply to the availability of these data, so they are not publicly available; however, they are available from the corresponding author upon reasonable request.

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
