# Peer review of "Serum Zinc and Long-Term Prognosis after Acute Traumatic Brain Injury with Intracranial Injury: A Multicenter Prospective Study†"

_jcm, 2022, doi:10.3390/jcm11216496_

Round 1

Reviewer 1 Report

Kim et al report on serum zinc levels prognostic value in TBI with intracranial injury. The report is based on a prospective observational study NCT04718935 (ongoing) for which primary outcomes measures are survival of TBI patients at 6 months, including cause of death, functional outcome of TBI assessed by GOS and quality of life.

Although zinc measurements and stratification by categorical levels are not mentioned in the NCT, it appears that this is a data mining study interim analysis with no pre-determined hypothesis and the results of subgroup analysis appear to be findings not hypothesized in advance. As such more stringent statistics should be employed such as Bonferroni adjustments with lower p values than traditional alpha of 0.05.

Reference 20 is a report on Vitamin D Deficiency and Prognosis after Traumatic Brain Injury with Intracranial Injury on the same population (??) and determining it as a prognostic factor similar to what is concluded for zinc in this manuscript (some of the same authors). It is not a report on the PATOS-TBI study. They concluded that Vitamin D deficiency is associated with poor functional outcomes at hospital discharge and mortality at 6-months after injury in TBI patients with intracranial hemorrhage or diffuse axonal injury. Rather than analyzing each variable separately, a predictive model including all potential variable should be done.

The authors have not demonstrated if zinc is an independent predictor nor did they calculated sensitivity, specificity etc…for the various cutoffs in zinc levels.

1.    The method does not detail how zinc was measured. Is it free zinc or includes protein bound?

2.    Who did the phone interviews? Trained? What questions were asked? How many interviewers?

3.    Low, low normal had higher ISS and therefore not isolated TBI which casts doubts about the isolated contribution of TBI on zinc levels.

4.    “S100B protein could be in the causative path of free serum zinc levels. neuronal S100B, a calcium and zinc binding damage-associated molecular pattern (DAMP), whose chronic upregulation is associated with aging, Alzheimer's disease (AD), motor neuron disease and traumatic brain injury (TBI).  S100B acts as sensor and regulator of elevated zinc levels in the brain and this metal-buffering activity is tied to a neuroprotective role.

While most of the total brain zinc exists in a protein-bound state, “free” zinc ions can be found predominantly within synaptic vesicles (Palmiter et al., 1996; Cole et al., 1999), where zinc gets released together with glutamate from glutamatergic presynaptic terminals. Whether high levels of S100B can affect local zinc concentrations by scavenging free zinc ions, which may lower toxic effects of zinc in situations of high zinc release such as over-excitation of glutamatergic neurons”.

5.    There is now mounting evidence suggesting that tau phosphorylation may be regulated by metal ions (such as iron, zinc and copper), which themselves are implicated in neurodegenerative disorders such as TBI

6.    After primary brain injury, translocated free zinc can accumulate in neurons and lead to secondary events such as oxidative stress, inflammation, edema, swelling, and cognitive impairment.

7.    TBI-induces progenitor cell proliferation and neurogenesis, suggesting that zinc has an essential role for modulating hippocampal neurogenesis after TBI.

8.    Zinc may reduce TBI-associated depression, a common and difficult outcome to treat in all forms of TBI.

Many reviews and reports address the potential importance of zinc and other metals in TBI. Although this study is interesting, the comments above should be considered by the authors.

Author Response

Reviewer 1

Kim et al report on serum zinc levels prognostic value in TBI with intracranial injury. The report is based on a prospective observational study NCT04718935 (ongoing) for which primary outcomes measures are survival of TBI patients at 6 months, including cause of death, functional outcome of TBI assessed by GOS and quality of life.

(ANSWER) Thank you for the review and the valuable comments. Each comment has been addressed.

Although zinc measurements and stratification by categorical levels are not mentioned in the NCT, it appears that this is a data mining study interim analysis with no pre-determined hypothesis and the results of subgroup analysis appear to be findings not hypothesized in advance. As such more stringent statistics should be employed such as Bonferroni adjustments with lower p values than traditional alpha of 0.05.

(ANSWER) Thank you for your comments. I added Bonferroni adjustments for post-hoc analysis of chi-square test. I revised the Methods section and Table 1 accordingly.

(REVISION: Methods)

A p-value under 0.05 was considered as the measure of statistical significance with Bonferroni adjustment for multiple comparisons.

(REVISION: Table 1)

Reference 20 is a report on Vitamin D Deficiency and Prognosis after Traumatic Brain Injury with Intracranial Injury on the same population (??) and determining it as a prognostic factor similar to what is concluded for zinc in this manuscript (some of the same authors). It is not a report on the PATOS-TBI study. They concluded that Vitamin D deficiency is associated with poor functional outcomes at hospital discharge and mortality at 6-months after injury in TBI patients with intracranial hemorrhage or diffuse axonal injury. Rather than analyzing each variable separately, a predictive model including all potential variable should be done.

(ANSWER) Thank you for your comments. I revised the Methods section accordingly. Following your opinion, we will develop a predictive model including all potential variables in the future.

(REVISION: Methods)

This was a multi-center prospective cohort study conducted in five participating academic hospitals in Korea based on the Pan-Asian Trauma Outcome Study for Traumatic Brain Injury (PATOS-TBI) registry (ClinicalTrials.gov, ID: NCT04718935).  

The authors have not demonstrated if zinc is an independent predictor nor did they calculated sensitivity, specificity etc…for the various cutoffs in zinc levels.

(ANSWER) Thank you for your comments. I added the independent predictive performances (accuracy, sensitivity, specificity, PPV, and NPV) of serum zinc levels on long- and short-term neurological prognoses accordingly. The performance was moderate good (Accuracy 0.7618 for 6-month mortality).

(REVISION: Methods)

Independent predictive performances of serum zinc levels on long- and short-term neurological prognoses was evaluated and sensitivity for 6-month mortality by low serum zinc level (~79.9 mcg/dl) was 0.9271 (Supplementary Table S1).

(REVISION: Supplementary Table S1)

  1. The method does not detail how zinc was measured. Is it free zinc or includes protein bound?

(ANSWER) Thank you for your comments. I revised the Methods section accordingly.

(REVISION: Methods)

Inductively coupled plasma mass spectrometry (ICP-MS) was used to measure zinc, including free zinc and protein-binding form, in metal free serum.

  1. Who did the phone interviews? Trained? What questions were asked? How many interviewers?

(ANSWER) Thank you for your comments. We trained the 5 research coordinators to evaluate patients’ functional outcome (GOS, EQ-5D) at 1 and 6 months after injury. The interviewer first asks if the patient is alive and if he or she has a disability (Vegetative, Severe, Moderate, Mild). After confirming whether to return to work, the questions about EQ-5D are as follows.

  • Is the patient having trouble exercising?
  • Does the patient have problems with self-management?
  • Does the patient have problems with your daily activities?
  • Does the patient have any pain or discomfort?
  • Does the patient have any anxiety or depression?

 I revised the Methods section accordingly.

(REVISION: Methods)

Long-term functional outcomes were investigated and collected by telephone interviews at 1 and 6 months after injury. For quality assurance of the PATOS-TBI registry, research coordinators were required to receive education and training prior to participation in the study and periodically during the study period. Quality control of entered data was performed by the PATOS-TBI data quality control team on a monthly basis.

(REVISION: Methods)

Research coordinators in each five participating hospitals interviewed with structured questions to assess the functional outcomes (GOS, EQ-5D, and reason of mortality) of patients at 1 and 6 months after injury.

  1. Low, low normal had higher ISS and therefore not isolated TBI which casts doubts about the isolated contribution of TBI on zinc levels.

(ANSWER) Thank you for your comments. As you pointed out, the ISS probably contributed to the patient's neurological prognosis. Because the study was not limited to isolated TBI patients, it is possible that severe injuries to anatomical sites other than the brain affected the patient's prognosis. Although head AIS was adjusted in the final statistical model, uncontrolled bias may occur. I revised the Discussion - limitations section accordingly.

(REVISION: Discussion - limitations)

Fourth, since the study population was not limited to isolated TBI patients, there is a possibility that critical injuries in anatomical regions other than the brain may have affected the prognosis of the patients. Although head AIS was adjusted in the final statistical model, uncontrolled bias may occur.  

  1. “S100B protein could be in the causative path of free serum zinc levels. neuronal S100B, a calcium and zinc binding damage-associated molecular pattern (DAMP), whose chronic upregulation is associated with aging, Alzheimer's disease (AD), motor neuron disease and traumatic brain injury (TBI). S100B acts as sensor and regulator of elevated zinc levels in the brain and this metal-buffering activity is tied to a neuroprotective role.

While most of the total brain zinc exists in a protein-bound state, “free” zinc ions can be found predominantly within synaptic vesicles (Palmiter et al., 1996; Cole et al., 1999), where zinc gets released together with glutamate from glutamatergic presynaptic terminals. Whether high levels of S100B can affect local zinc concentrations by scavenging free zinc ions, which may lower toxic effects of zinc in situations of high zinc release such as over-excitation of glutamatergic neurons”.

  1. There is now mounting evidence suggesting that tau phosphorylation may be regulated by metal ions (such as iron, zinc and copper), which themselves are implicated in neurodegenerative disorders such as TBI
  2. After primary brain injury, translocated free zinc can accumulate in neurons and lead to secondary events such as oxidative stress, inflammation, edema, swelling, and cognitive impairment.

(ANSWER) Thank you for your comments. Because zinc ions have both neuroprotective and neurotoxic effects, it has been reported that abnormalities in zinc homeostasis in cells are associated with neurodegeneration and neurological disorders. The extracellular release of free zinc ions is known to be regulated by zinc binding proteins such as S100B. I reviewed the related articles and added the Discussion section accordingly.

(REVISION: Discussion)

Alterations in free zinc ion homeostasis, such as accumulation in neurons, are known to impair the advanced brain functions including memory and cognition.[29] S100B, one of zinc binding proteins, affects local zinc concentrations by scavenging free zinc ions from extracellular release and regulates hyperexcitation of glutamatergic neurons in the central nervous system [30]. It has been demonstrated that zinc-bound S100B has the potential to modulate excitotoxicity by altering calcium signaling, which could be reflected in serum zinc levels.[31] Moreover, zinc plays a role in regulating tau phosphorylation, which may contribute to pathological structural changes in the TBI, and zinc supplementation is suggested to decrease tau phosphorylation by negatively regulating the amount of GSK-3-beta.[32,33] Considering that serum zinc level is physiologically decreased during the acute phase of TBI, the failure of the tau phosphorylation role in response to dynamic changes in zinc amount may be associated with poor prognosis.

  1. Lovell, M.A.; Xie, C.; Markesbery, W.R. Protection against amyloid beta peptide toxicity by zinc. Brain research 1999, 823, 88-95, doi:10.1016/s0006-8993(99)01114-2.
  2. Hagmeyer, S.; Cristóvão, J.S.; Mulvihill, J.J.E.; Boeckers, T.M.; Gomes, C.M.; Grabrucker, A.M. Zinc Binding to S100B Affords Regulation of Trace Metal Homeostasis and Excitotoxicity in the Brain. Frontiers in Molecular Neuroscience 2018, 10, doi:10.3389/fnmol.2017.00456.
  3. Foote, J.W.; Delves, H.T. Albumin bound and alpha 2-macroglobulin bound zinc concentrations in the sera of healthy adults. J Clin Pathol 1984, 37, 1050-1054, doi:10.1136/jcp.37.9.1050.
  4. Foote, J.W.; Delves, H.T. Albumin bound and alpha 2-macroglobulin bound zinc concentrations in the sera of healthy adults. J Clin Pathol 1984, 37, 1050-1054, doi:10.1136/jcp.37.9.1050.
  5. Yang, W.J.; Chen, W.; Chen, L.; Guo, Y.J.; Zeng, J.S.; Li, G.Y.; Tong, W.S. Involvement of tau phosphorylation in traumatic brain injury patients. Acta Neurol Scand 2017, 135, 622-627, doi:10.1111/ane.12644.

  1. TBI-induces progenitor cell proliferation and neurogenesis, suggesting that zinc has an essential role for modulating hippocampal neurogenesis after TBI.

(ANSWER) Thank you for your comments. The mechanism you point out (progenitor cell proliferation induction) is important to the theological backgrounds of this study. I revised the Introduction section accordingly.

(REVISION: Introduction)

With induction of progenitor cell proliferation and neurogenesis, zinc is also essential in the modulation of hippocampal neurogenesis after TBI.[12]  

  1. Zinc may reduce TBI-associated depression, a common and difficult outcome to treat in all forms of TBI.

(ANSWER) Thank you for your comments. I revised the Introduction section accordingly.

(REVISION: Introduction)

Moreover, it has been suggested that zinc supplementation given as a treatment after TBI can effectively reduce cognitive impairment and depression associated with TBI.[17-19]

  1. Levenson, C.W. Zinc and Traumatic Brain Injury: From Chelation to Supplementation. Med Sci (Basel) 2020, 8, doi:10.3390/medsci8030036.
  2. Cope, E.C.; Morris, D.R.; Levenson, C.W. Improving treatments and outcomes: an emerging role for zinc in traumatic brain injury. Nutr Rev 2012, 70, 410-413, doi:10.1111/j.1753-4887.2012.00486.x.
  3. Cope, E.C.; Morris, D.R.; Scrimgeour, A.G.; Levenson, C.W. Use of zinc as a treatment for traumatic brain injury in the rat: effects on cognitive and behavioral outcomes. Neurorehabil Neural Repair 2012, 26, 907-913, doi:10.1177/1545968311435337.

Many reviews and reports address the potential importance of zinc and other metals in TBI. Although this study is interesting, the comments above should be considered by the authors.

(ANSWER) Thank you for your comments.

Reviewer 2 Report

Reviewers read an article on a multicenter prospective study of serum zinc and long-term prognosis after acute trauma. Unfortunately, we assess the lack of scientific evidence to accept this paper with the conclusion that "zinc may be a biomarker in the acute phase of TBI" at the current data stage of the prospective study. The reviewers and authors seem to have different views on this paper.

Author are entitled to state a possibility, but to accept it as an academic paper, it is not science without sufficient evidence. Even reading the way the authors phrased their paper, mention the ”possibility” and then discuss the mechanism. First, scientific papers generally begin with a clear scientific result. Next, then make inferences about the mechanism. From this perspective, the reviewer evaluates that the paper uses multivariate analysis and logistic analysis to make arguments, but further data accumulation is needed to provide evidence to prove the conclusions.

However, if the data that can be said to be moderately reliable from this paper would be Table 4. The reviewer also suggests as an option to revise this article to read, "Zinc levels are related to TBI outcomes in a population with a background of diabetes mellitus.
But in cases such as fulminant type 1 diabetes, the zinc transporter, the diabetes caused by a single nucleotide substitution in the ZnT8 gene has proven to be closely related to zinc levels and neurological sequelae, but this discussion of the condition is not at all applicable to normal diabetes. If it were to be cited in TBI, it would only be of reference value.

If such a research model is to be promoted, it should be based on the use of two groups of animal model mice of adjusted with normal development, uniform age and body size. And differing zinc levels, with both groups developing comparable TBI, and the subsequent neurological prognosis must be studied under the basic sciences. So reviewer recommend that you verify and cite from the latest animal study data, or change the content of your submission to fully cite such a paper, before resubmitting it to JCM.

Best regards,

Dr. Reviewer 

Author Response

Reviewer 2

Reviewers read an article on a multicenter prospective study of serum zinc and long-term prognosis after acute trauma. Unfortunately, we assess the lack of scientific evidence to accept this paper with the conclusion that "zinc may be a biomarker in the acute phase of TBI" at the current data stage of the prospective study. The reviewers and authors seem to have different views on this paper.

Author are entitled to state a possibility, but to accept it as an academic paper, it is not science without sufficient evidence. Even reading the way the authors phrased their paper, mention the ”possibility” and then discuss the mechanism. First, scientific papers generally begin with a clear scientific result. Next, then make inferences about the mechanism. From this perspective, the reviewer evaluates that the paper uses multivariate analysis and logistic analysis to make arguments, but further data accumulation is needed to provide evidence to prove the conclusions.

(ANSWER) Thank you for the review and the valuable comments. Each comment has been addressed. I deleted the description of possibility of zinc as a biomarker for TBI in the Discussion and Conclusions sections accordingly.

(REVISION: Conclusions)

Serum zinc deficiency is associated with long- and short-term mortality and poor functional recovery for TBI patients with intracranial hemorrhage and diffuse axonal injury. Furthermore, these associations are strengthened in the TBI patients previously diagnosed with diabetes mellitus.

However, if the data that can be said to be moderately reliable from this paper would be Table 4. The reviewer also suggests as an option to revise this article to read, "Zinc levels are related to TBI outcomes in a population with a background of diabetes mellitus.

But in cases such as fulminant type 1 diabetes, the zinc transporter, the diabetes caused by a single nucleotide substitution in the ZnT8 gene has proven to be closely related to zinc levels and neurological sequelae, but this discussion of the condition is not at all applicable to normal diabetes. If it were to be cited in TBI, it would only be of reference value.

(ANSWER) Thank you for your comments. I reviewed the related articles and added the Discussion section accordingly.

(REVISION: Discussion)

Serum zinc levels in patients with type 1 and type 2 diabetes are known to be lower than non-diabetic patients.[35,36] Zinc supplements are recommended for diabetic patients due to their antioxidant effects on multiple organs, such as coronary heart disease.[37] Since zinc would be released from liver and be excreted through kidney after TBI,[16] it can be assumed that zinc levels in the ED indicated not only general serum zinc level, but total storage of zinc. Considering this association, serum zinc level of patients with diabetes mellitus should be managed to improve neurological outcomes after TBI with intracranial injury.

  1. Masood, N.; Baloch, G.H.; Ghori, R.A.; Memon, I.A.; Memon, M.A.; Memon, M.S. Serum zinc and magnesium in type-2 diabetic patients. J Coll Physicians Surg Pak 2009, 19, 483-486, doi:08.2009/JCPSP.483486.
  2. Lin, C.C.; Huang, Y.L. Chromium, zinc and magnesium status in type 1 diabetes. Curr Opin Clin Nutr Metab Care 2015, 18, 588-592, doi:10.1097/MCO.0000000000000225.
  3. Hamedifard, Z.; Farrokhian, A.; Reiner, Z.; Bahmani, F.; Asemi, Z.; Ghotbi, M.; Taghizadeh, M. The effects of combined magnesium and zinc supplementation on metabolic status in patients with type 2 diabetes mellitus and coronary heart disease. Lipids Health Dis 2020, 19, 112, doi:10.1186/s12944-020-01298-4.

If such a research model is to be promoted, it should be based on the use of two groups of animal model mice of adjusted with normal development, uniform age and body size. And differing zinc levels, with both groups developing comparable TBI, and the subsequent neurological prognosis must be studied under the basic sciences. So reviewer recommend that you verify and cite from the latest animal study data, or change the content of your submission to fully cite such a paper, before resubmitting it to JCM.

(ANSWER) Thank you for your comments. I reviewed the related articles and added the Discussion section accordingly.

(REVISION: Discussion)

In recent animal studies, zinc supplement showed improvements in spatial learning and memory, and hippocampal neuronal precursor cell regenesis.[19,28] Neuronal precursor cell proliferation and neurogenesis were found in the hippocampus after TBI,[28] while zinc administration shortly after TBI had demonstrated enhanced protein synthesis and improved consciousness.[27]

  1. Cope, E.C.; Morris, D.R.; Scrimgeour, A.G.; Levenson, C.W. Use of zinc as a treatment for traumatic brain injury in the rat: effects on cognitive and behavioral outcomes. Neurorehabil Neural Repair 2012, 26, 907-913, doi:10.1177/1545968311435337.
  2. Young, B.; Ott, L.; Kasarskis, E.; Rapp, R.; Moles, K.; Dempsey, R.J.; Tibbs, P.A.; Kryscio, R.; McClain, C. Zinc supplementation is associated with improved neurologic recovery rate and visceral protein levels of patients with severe closed head injury. J Neurotrauma 1996, 13, 25-34, doi:10.1089/neu.1996.13.25.
  3. Cope, E.C.; Morris, D.R.; Gower-Winter, S.D.; Brownstein, N.C.; Levenson, C.W. Effect of zinc supplementation on neuronal precursor proliferation in the rat hippocampus after traumatic brain injury. Exp Neurol 2016, 279, 96-103, doi:10.1016/j.expneurol.2016.02.017.

Round 2

Reviewer 1 Report

I thank the authors for responding appropriately to my comments.

Author Response

I thank the authors for responding appropriately to my comments.

(ANSWER) Thank you for the review and the valuable comments. I received English editing services for this manuscript.

Reviewer 2 Report

The authors responded to all of the reviewers' comments. The mechanism of the relationship between basic TBI and zinc was also revised with additional references. The literature was taken from Basic Science of Animal Models, and I believe this work upgraded the paper. There were some minor revisions to the methods and tables, so the authors should double-check that the final data is accurate.

Author Response

The authors responded to all of the reviewers' comments. The mechanism of the relationship between basic TBI and zinc was also revised with additional references. The literature was taken from Basic Science of Animal Models, and I believe this work upgraded the paper. There were some minor revisions to the methods and tables, so the authors should double-check that the final data is accurate.

(ANSWER) Thank you for the review and the valuable comments. I checked the manuscript accordingly.